# Validation of an Analytical Method of 3′,4′,5-Trihydroxy-3-Methoxy-6,7-Methylenedioxyflavone 4′-Glucuronide for Standardization of *Spinacia oleracea*

**DOI:** 10.3390/molecules29112494

**Published:** 2024-05-24

**Authors:** Yun Gon Son, Juyoung Jung, Dong Kun Lee, Sang Won Park, Jeong Yoon Kim, Hyun Joon Kim

**Affiliations:** 1Department of Pharmaceutical Engineering, Institute of Agricultural and Life Science, Anti-aging Bio Cell Factory Regional Leading Research Center, Gyeongsang National University, Jinju 52725, Republic of Korea; thsdbsrhs112@naver.com; 2Brain Health Lab. Co., Ltd., Business Incubating Center (C-203), Gyeongsang National University, 501 Jinju-daero, Jinju 52828, Republic of Korea; juyoung.jung@gmail.com; 3Tyrosine Peptide Multiuse Research Group, Department of Physiology and Convergence Medical Sciences, Institute of Medical Sciences, College of Medicine, Antiaging Bio Cell Factory Regional Leading Research Center, Gyeongsang National University, 15 Jinju-daero 816 Beongil, Jinju 52727, Republic of Korea; dklee@gnu.ac.kr; 4Tyrosine Peptide Multiuse Research Group, Department of Pharmacology and Convergence Medical Sciences, Institute of Medical Sciences, College of Medicine, Antiaging Bio Cell Factory Regional Leading Research Center, Gyeongsang National University, 15 Jinju-daero 816 Beongil, Jinju 52727, Republic of Korea; parksw@gnu.ac.kr; 5Tyrosine Peptide Multiuse Research Group, Department of Anatomy and Convergence Medical Sciences, Institute of Medical Sciences, College of Medicine, Antiaging Bio Cell Factory Regional Leading Research Center, Gyeongsang National University, 15 Jinju-daero 816 Beongil, Jinju 52727, Republic of Korea

**Keywords:** spinach, HPLC method validation, quantitative analysis, nutraceuticals, TMG

## Abstract

Spinach (*Spinacia oleracea*) is one of the most famous vegetables worldwide, rich in essential metabolites for various health benefits. It is a valuable plant source that has the potential to be a nutraceutical. This study aimed to evaluate the single characteristic marker compound to establish the validation of HPLC-DAD methods applied to the development of a nutraceutical using spinach samples. Six metabolites (**1**–**6**) were identified from the spinach samples such as freeze-dried spinach (FDS) and spinach extract concentrate (SEC) by LC-Q-TOF/MS analysis. Among the six metabolites, 3′,4′,5-trihydroxy-3-methoxy-6,7-methylenedioxyflavone 4′-glucuronide (TMG) was selected as a marker compound due to its highest abundance and high selectivity. The specificity, accuracy, linearity, precision, repeatability, limit of detection (LOD), and limit of quantification (LOQ) of TMG in the spinach samples (FDS and SEC) were validated according to AOAC international guideline. The specificity was confirmed by monitoring the well separation of the marker compound from other compounds of spinach samples in the base peak intensity (BPI) and ultraviolet (UV) chromatogram. The calibration curve of TMG (15.625~500 μg/mL) had reasonable linearity (R^2^ = 0.999) considered with LOD and LOQ values, respectively. Recovery rate of TMG was 93–101% for FDS and 90–95% for SEC. The precision was less than 3 and 6% in the intraday and interday. As a result, the HPLC-DAD validation method of TMG in the spinach samples (FDS and SEC) was first established with AOAC and KFDA regulations for approving functional ingredients in functional foods.

## 1. Introduction

Spinach (*Spinacia oleracea*) is an annual plant belonging to the Amaranthaceae family [1]. It is widely cultivated as a green leafy versatile vegetable for various cuisines, including fresh ones, soups, sauces, etc. [2]. Spinach is widely recognized for its nutritional value, containing sugars, organic acids, amino acids, vitamins, minerals, and phytochemicals [3,4,5]. Recent studies have highlighted the association between phytochemicals found in spinach and several health benefits [6]. Spinach has been found to have positive impacts on eye health, as well as anti-inflammatory and antioxidant effects [7,8,9]. Moreover, it has also been linked to the prevention and management of chronic diseases such as depression, obesity, and hypolipidemia [10]. With its impressive benefits, spinach has valuable potential for development as a nutraceutical.

The healthy benefits of spinach have been noted to involve its specific phytochemicals, such as carotenoids, flavonoids, terpenoids, polyketides, lignans, saponins, and peptides [11,12,13]. One of the key components responsible for these benefits is flavonoids, a type of phytochemical in spinach [14,15]. The types of spinach flavonoids have been known as patuletin derivatives, spinacetin derivatives, spinach flavonoids glucuronide derivatives, and other basic skeleton flavonoids [16,17,18]. One of the patuletin derivatives, patuletin-3-O-(2′′-coumaroylglucosyl)-(1→6)-[apiosyl-(1→2)]-β-D-glucopyranoside, has been reported to inhibit the formation of advanced glycation end products (AGEs) and aldose reductase activity (RLAR) [19]. 3′,4′,5-Trihydroxy-3-methoxy-6,7-methylenedioxyflavone 4′-glucuronide (TMG) is a specific flavonoid glucuronide derivative found in spinach leaves [20,21,22]. TMG has inhibitory activity on granulation in basophilic leukemia RBL-2G3 cells [23]. The results indicate that spinach flavonoids have therapeutic potential and specific skeletons for standard compounds of spinach.

Spinach is a well-known vegetable that has high biological activities to be applied as a functional food. To develop the substance for functional foods, it is very important to discover the reasonable marker compound that can reveal the identity of spinach to set the validation of the HPLC analysis method [24]. These marker compounds serve as indicators or reference points for quality control and standardization of nutraceutical products [25]. The selection of standard compounds is typically based on their presence in the botanical source or their known bioactivity [26,27]. The Ministry of Food and Drug Safety’s regulations include testing methods for functional ingredients of health functional foods in ‘Standards and Specifications for Health Functional Foods’, ‘Standards and Specifications for Foods’, and ‘Standards and Specifications for Food Additives’, and it is stated that methods are approved by the Codex Alimentarius Commission (CAC) and AOAC methods. If there is no established test method for the raw functional material, the validity of the test method using a characteristic marker compound should be presented including the items (specificity, accuracy, precision, quantitation limit, linearity, and range) [28].

We aimed to strictly adhere to international standard guidelines and present an effective test method for spinach analysis. 3′,4′,5-Trihydroxy-3-methoxy-6,7-methylenedioxyflavone 4′-glucuronide (TMG) was firstly identified as a marker compound to develop nutraceuticals of spinach. In addition, the effectiveness of the analytical method validation was verified by confirming specificity, precision, repeatability, accuracy, limit of detection (LOD), limit of quantitation (LOQ), and recovery rate according to the AOAC international guideline. 

## 2. Results and Discussion

### 2.1. Metabolite Identification of Spinach Extract Using LC-Q-TOF/MS

The identification of metabolites from the spinach was conducted using LC-Q-TOF/MS analysis with positive mode. The six peaks were annotated from the base peak intensity (BPI) per gram of freeze-dried spinach (FDS) extract (Figure 1A). Additionally, Figure 1B displayed a UV chromatogram at 350 nm, which corresponded to the high sensitivity of the six target metabolites selectively. The six peaks were separated with sufficient sensitivity, retention time (*t*_R_), and resolution. The mass spectra for the six peaks were compared with previous data [18,20]. The six compounds were flavonoid derivatives such as spinacetin 3-*O*-glucosyl-(1→6)-[apiosyl(1→2)]-glucoside (**1**), spinacetin-3-*O*-glucosyl-(1→6)-glucoside (**2**), spinatoside (**3**), jaceidin 4′-glucuronide (**4**), 3′,4′,5-trihydroxy-3-methoxy-6,7-methylenedioxyflavone 4′-glucuronide (**5**), and 5,4′-dihydroxy-3,3′-dimethoxy-6,7-methylendioxyflavone-4′-glucuronide (**6**) through positive ion mode by comparison with relevant references [18,20].

Peak 1 (*t*_R_ = 11.61 min, Figure 1C) was observed as the molecular ion peak at *m*/*z* 803.2242 and two fragment ion peaks at *m*/*z* 671.1843 and 347.0771. The fragment ion at *m*/*z* 671.1843 ([M + H]^+^—132 amu) was derived from its apioside moiety cleavage. Furthermore, the two glucose moieties were fragmented to obtain the ion peak at *m*/*z* 347.0771 ([M + H]^+^—324 amu). Based on this typical fragment pattern, peak 1 was identified as spinacetin 3-O-glucosyl-(1→6)-[apiosyl(1→2)]-glucoside. Peak 2 (*t*_R_ = 12.72 min, Figure 1D) exhibited a molecular ion peak at *m*/*z* 671.1818 ([M + H]^+^) with the fragment ion observed at *m*/*z* 347.0769 ([M + H]^+^—324 amu) corresponding to the loss of two glucose moieties. Based on these mass data, it was tentatively assigned as spinacetin 3-*O*-glucosyl-(1→6)-glucoside. Peaks 3–6 only showed the molecular ions, which were compared to the theoretical value to presume the metabolite with a reasonable error value (Table 1). Peak 3 (*t*_R_ = 14.96 min, Figure 1E) was assigned as spinatoside, which had the molecular ion peak at *m*/*z* 523.1066 with −4.18 ppm of error value by comparing the theoretical mass at *m*/*z* 523.1087. Peak 4 (*t*_R_ = 15.87 min, Figure 1F) displayed the molecular ion peak at *m*/*z* 537.1240, corresponding to the theoretical mass of *m*/*z* 537.1240 with an error value of −0.81 ppm to identify jaceidin 4′-glucuronide. Peak 5 (*t*_R_ = 17.69 min, Figure 1G) was confirmed as −1.99 ppm of an error value. In accordance, it was annotated as 3′,4′,5-trihydroxy-3-methoxy-6,7-methylenedioxyflavone 4′-glucuronide with the chemical formula C_23_H_20_O_14_. Peak 6 (*t*_R_ = 18.62 min, Figure 1H) was observed as a molecular ion peak at *m*/*z* 535.1068 with −5.2 ppm of an error. It was in agreement with 5,4′-dihydroxy-3,3′-dimethoxy-6,7-methylendioxyflavone-4′-glucuronide. 

### 2.2. Establishment of TMG as a Standard Compound of Spinach

Based on LC-Q-TOF/MS analysis, it was confirmed that spinach contained a high level of six flavonoid derivatives. Among them, TMG was selected as a standard compound for establishing the quantitative validation method of spinach. TMG is the most easily purchasable authentic compound of the identified six spinach metabolites. It also had the highest content on both the chromatogram of UV and BPI by UPLC-Q-TOF/MS. Thus, TMG was selected as a marker compound of spinach in this study. Moreover, the molecular extinction coefficient of TMG was presented (Appendix A, Appendix A).

### 2.3. Development HPLC Validation Method for Spinach Sample Using Standard Compound

#### 2.3.1. Specificity

As shown in Appendix A, the various wavelength and analytical methods were performed to select the best selectivity conditions. HPLC analysis was conducted at a wavelength of 350 nm, where the identified six flavonoid derivatives were selectively detected with high levels. As shown in Figure 2B,C, HPLC chromatogram of the sample solution (FDS and SEC) confirmed that the metabolites were properly separated. Figure 2A shows the retention time (*t*_R_ = 31.459 min) of TMG through established HPLC methods. Furthermore, the location of TMG in the sample solution (FDS and SEC) was accurately characterized to detect without the influence of interfering substances (Appendix A). Therefore, these findings supported specificity of TMG as a standard compound through the verified HPLC methods.

#### 2.3.2. Linearity

TMG standard solutions in the range 15.625, 31.25, 62.5, 125, 250, and 500 μg/mL were prepared and analyzed under the established HPLC conditions to obtain peak areas at each concentration. A standard curve was created using TMG concentration on the x-axis and peak area on the y-axis. The TMG regression line was drawn using experimental values for the standard curves with more than three repetitions. The linearity results of the ANOVA test for TMG concentration were confirmed (Appendix A). All three lines had 0.999 of the coefficient values (R^2^), representing a significant linearity. The linearity was also confirmed doubly with the residual plot of TMG to determine the appropriate linear model (Appendix A).

#### 2.3.3. Limit of Detection (LOD) and Limit of Quantification (LOQ)

LOD and LOQ were determined using the TMG regression line of the standard curves that were also used for linear verification. To calculate the LOD and LOQ, we obtained the slope (S) and intercept (σ) values through the regression equation derived from the regression line. The results of three experiments showed the mean slope (S) at 0.1464 and the standard deviation of the intercept (σ) at 0.0074. Using these values in the proper equation, LOD and LOQ of TMG were calculated to be 0.167 μg/mL and 0.505 μg/mL, respectively (Table 2).

#### 2.3.4. Accuracy

The recovery rate was confirmed by adding TMG (standard compound) corresponding to the concentrations at 50%, 100%, and 150% to TMG-containing sample solutions (FDS and SEC). In the FDS sample, the concentration of TMG was 17.97 μg/mL. Thus, TMG was added to the FDS sample with 8.98, 17.97, and 26.95 μg/mL corresponding to 50%, 100%, and 150% of this content, respectively. The average recovery rate for TMG in FDS samples was 97%, with a range of 93% to 101% (Table 3). The TMG concentration in the SEC sample solution was 41.75 μg/mL. To achieve 50%, 100%, and 150% of the TMG content, the SEC sample was supplemented with 20.88, 41.75, and 62.63 μg/mL of TMG, respectively. As a result, the average and range of recovery rate exhibited 93% and 90~95%, respectively (Table 4). The accuracy of the recovery test complied with the standard ranges of 90–108% at a concentration of 0.1% (1 mg/g) according to AOAC international accuracy and KFDA guidelines for functional ingredient approval in health functional foods.

#### 2.3.5. Precision

##### Intraday Precision (Repeatability)

Repeated measurements (*n* = 5) were taken with 0.5, 1.0, and 1.5 g of FDS and SEC to confirm repeatability with respect to the sample amount. For intraday precision for FDS, 0.5, 1.0, and 1.5 g had a mean of TMG contents with 1043, 996, and 1090 μg/g for 5 times repeated experiments. As shown in Table 5, the relative standard deviation (RSD, %) was 1.9, 2.2, and 0.3 for each FDS amount, respectively. In intraday precision test for SEC, the mean TMG contents for 0.5 g, 1.0 g, and 1.5 g were 2576, 2466, and 2475 µg/g, respectively, based on five repeated experiments. Table 6 shows that the RSD (%) varied 1.3, 0.9, and 1.7 for 0.5, 1.0, and 1.5 g of SEC amounts. The intraday results of the ANOVA test for TMG contents at each FDS and SEC amount were confirmed (Appendix A). Intraday precision confirmed the appropriate RSD level (<3%) at the concentration of 0.1% (1 mg/g) based on AOAC international and KFDA guidelines [28,29].

##### Interday Precision (Intermediate Precision)

To ensure intraday precision, the repeated experiments were conducted five times a day for five consecutive days. The mean of TMG content exhibited 1111 and 1091 μg/g in 0.5 g and 1.5 g of FDS, respectively. The RSD for both weights was 0.7% and 0.6%, respectively (Table 7). The mean of TMG content in 0.5 g of SEC was 2603 μg/g, while, in 1.5 g of SEC, it was 2587 μg/g. The RSD for both weights was 1.4% and 1.9%, respectively (Table 8). According to the guidelines, intermediate precision was achieved as the pooled RSD was less than 6% based on a concentration of 0.1% (1 mg/g).

## 3. Materials and Methods

### 3.1. Plant Materials

The spinach was cultivated at a local farm in Goseong-gun, Gyeongsangnam-do, Republic of Korea. The seed of spinach was sown around September to October 2022. The aerial parts were harvested between November 2022 and March 2023, 60 days after sowing, to use for experiments. The harvested spinach was washed to remove the contaminants and then dried using a freeze dryer (30 L, IlshinBioBase Ltd., Dongducheon-si, Gyeonggi-do, Republic of Korea) for 48 h. The freeze-dried spinach (500 kg) was extracted with 70% ethanol to be six times the volume of the plant material. The filtered crude spinach extract was concentrated using an evaporator and dried using freeze-drying to obtain a powder type at a company equipped for extraction facilities (Kuan Industrial Co., Ltd. Factory, Geumsan-gun, Chungcheongnam-do, Republic of Korea). In this experiment, freeze-dried spinach (FDS) and spinach extract concentrate (SEC) were used for analysis.

### 3.2. LC-Q-TOF/MS Analysis

The metabolites from the spinach samples were annotated using liquid chromatography (NEXERA UHPLC, Shimadzu, Kyoto, Japan) connecting with quadrupole time of flight mass (Q-TOF/MS, AB SCIEX X500R, Framingham, MA, USA). To prepare the analytical sample, 1 g of freeze-dried spinach was sonicated with 50 mL of distilled water for 1 h. The spinach extract (1 μL) was injected to the column (Poroshell 120 EC C18, 2.7 μm, 2.1 × 100 mm, Agilent Technologies, Santa Clara, CA, USA). The eluent solvents comprised water (A) and acetonitrile containing 0.1% acetic acid (B). The gradient conditions were conducted for 40 min: 0 min, 0% B; 30 min, 50% B; and 40 min, 100% B. The MS ionization was positive mode using electrospray ionization (ESI) with 5500 V of ion spray voltage and 450 °C of ion source temperature. The gas condition was kept at 30 psi of curtain gas pressure, 50 psi nebulizer gas pressure, and 50 psi of heating gas pressure with 50 V decluttering potential. The MS scan range was 50–1000 *m*/*z* with a collision energy ramp of 10–30 eV (Appendix A).

### 3.3. Method Validation

The standard compound (TMG) of spinach was validated using a high-performance liquid chromatography–diode array detector (HPLC-DAD, Appendix A) following the guidelines of the Korea Food and Drug Administration (KFDA). The validation method comprised of five criteria including linearity, specificity, accuracy, precision with repeatability and reproducibility, and limit of detection and quantitation (LOD and LOQ). According to these guidelines, verification of the standard compound (TMG) analysis method was performed, along with quantification of TMG in the analysis sample (FDS and SEC). The experiments were validated through appropriate repetitions of each procedure.

### 3.4. Standard Solution Preparation Using TMG

3′,4′,5-Trihydroxy-3-methoxy-6,7-methylene-dioxyflavone 4′-glucuronide (TMG) as the standard compound of spinach sample was purchased from Molport, Inc. (Beacon, NY, USA). A total of 10 mg of TMG was weighed and dissolved in 10 mL of methanol to make a stock solution. A TMG standard solution was prepared by appropriately diluting the above standard stock solution with methanol to use for validation methods.

### 3.5. Sample Solution Preparation Using FDS and SEC

The 0.5, 1.0, and 1.5 g of each sample (FDS and SEC) were extracted with 20 mL of 80% ethanol using a sonicator for 90 min to prepare samples with concentrations of 25, 50, and 75 mg/mL. Each solution was filtered through 0.2 μm membrane filter (hydrophilic unit, Advantec MFD, Inc., Dublin, CA, USA) to use in accordance with the validation methods.

### 3.6. HPLC-DAD Analysis

The method validation was mainly performed using Agilent Technology Infinity 1260 HPLC equipped with an autosampler, binary pump, and diode array detector (DAD). The standard and sample solution (injection volume: 10 μL) were chromatographed on the column (XBridge C18, 4.6 × 150 mm, 5 μm, Waters Corporation, Milford, MA, USA). The elution solvents A and B were 0.1% acetic acid in water and 0.1% acetic acid in acetonitrile. The gradient mode of elution was set as follows: 0–5 min, 90% A; 5–13 min, 85% A; 13–40 min, 70% A; 40–50 min, 50% A; and 50–60 min, 0% A. The UV wavelength was 350 nm, which was selected due to the relatively high intensity of TMG.

### 3.7. Specificity

The specificity of the HPLC analysis was determined by comparing the results of the standard solution (TMG, 62.5 μg/mL) and sample solution (FDS and SEC, 50 mg/mL). The TMG peak was confirmed to be completely separated from other peaks present in the chromatogram of spinach samples based on its retention time and UV spectrum. The specificity of the standard and sample solutions was confirmed through triplicate analysis.

### 3.8. Linearity

The TMG stock solution (1.0 mg/mL) was diluted to prepare six concentrations: 15.625, 31.25, 62.5, 125, 250, and 500 μg/mL. Methanol was used as a dilution solvent for the blank. All the diluted TMG solutions were measured using HPLC. The peak area of each TMG concentration was derived from the HPLC chromatogram to obtain the calibration curves. The coefficient of determination (R^2^) was confirmed for calibration curves. The linearity of the results was demonstrated by performing triplicate measurements.

### 3.9. Limit of Detection (LOD) and Limit of Quantification (LOQ)

A regression line was generated from the TMG standard curve for linearity verification, which displayed the correlation between TMG concentrations (0, 15.625, 31.25, 62.5, 125, 250, and 500 μg/mL) and peak area. The limit of detection (LOD) and limit of quantification (LOQ) were determined using the average (S) of the slope of the regression line and the standard deviation (σ) of the y-intercept of the regression line. The equation for the calculation was as follows: LOD = 3.3 × σ/S, LOQ = 10 × σ/S.

### 3.10. Accuracy

The FDS and SEC (each 0.5 g) were extracted with 10 mL of 80% ethanol using a sonicator for 90 min. Subsequently, 100 μL of the blank (80% ethanol) or TMG solution corresponding to 50%, 100%, and 150% was mixed with 900 μL of each FDS and SEC extract. The blank content for FDS and SEC was 17.97 and 41.75 μg/mL, respectively. To achieve concentrations of 50%, 100%, and 150%, 8.98, 17.97, and 26.95 µg/mL of TMG samples were added to the FDS, while 20.88, 41.75, and 62.63 µg/mL were added to the SEC. The accuracy was proved by triplicates.

### 3.11. Precision

#### 3.11.1. Intraday Precision (Repeatability)

The accurately weighed 0.5, 1.0, and 1.5 g of FDS and SEC were sonicated with 20 mL of 80% ethanol for 90 min to obtain the sample solutions. The TMG contents were measured in each sample solution using HPLC analysis. All the units of contents were converted to μg/g according to the dilution factor. Intraday precision was performed by measuring the TMG contents per gram of FDS and SEC. The experiment was conducted five times in total to confirm the repeatability.

#### 3.11.2. Interday Precision (Intermediate Precision)

The accurately weighed 0.5 and 1.5 g of FDS and SEC were sonicated with 20 mL of 80% ethanol for 90 min to obtain the sample solutions. The measurement of TMG contents and unit conversion in sample solution were conducted in the same manner as the intraday precision methods. Interday precision was measured five times per day for a total of five days to confirm the intermediate precision.

### 3.12. Statistical Analysis

The sample groups were analyzed using one-way ANOVA with *p*  <  0.05 in SPSS 17.0 (SPSS Inc., Chicago, IL, USA) to determine variable significance for linearity, accuracy, and precision. All the experiments were conducted in more than triplicate or more.

## 4. Conclusions

Six flavonoid glycosides (**1**–**6**) were identified from spinach extract using LC-Q-TOF/MS. Two sample types were prepared for effective nutraceutical development: freeze-dried spinach (FDS) and spinach extract concentrate (SEC). TMG with high specificity was selected as a standard compound of spinach samples to validate the HPLC analytical method according to AOAC international guidelines for linearity, LOD, LOQ, accuracy, and precision. The linearity of the TMG calibration curve from 15.625 to 500 μg/mL was verified with a high coefficient factor of 0.999. LOD and LOQ values were determined to be 0.167 and 0.505 μg/mL. The accuracy of the validation method was also confirmed to be in an average recovery rate range of 97% for FDS and 93% for SEC. The precision was also evaluated as appropriate for both intraday and interday RSD levels. The repeatability of the analysis was within the acceptable range of 0.3–2.2% for FDS and 0.9–1.7% for SEC, based on the amount at 0.5, 1.0, and 1.5 g. Additionally, the intermediate precision was determined to be highly reliable for each spinach sample. These results showed that the method validation using TMG as a standard compound for spinach was reliable, proven according to the AOAC international guidelines.

## Figures and Tables

**Figure 1 molecules-29-02494-f001:**
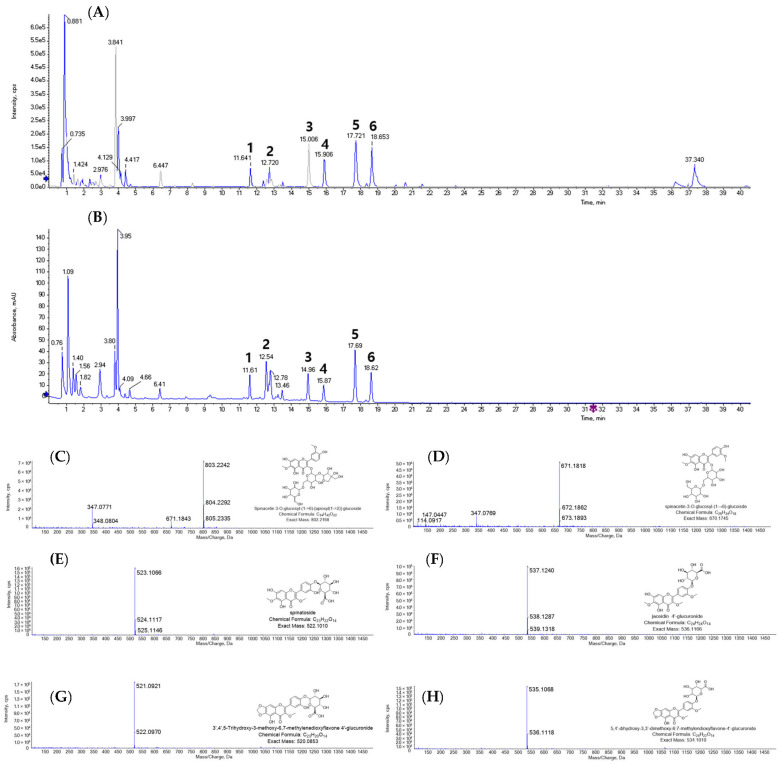
LC-Q-TOF/MS analysis of 80% ethanol extract of freeze-dried spinach (FDS). (**A**) Base peak intensity (BPI) and (**B**) ultraviolet (UV) chromatogram of FDS 80% ethanol extract. Individual mass gram of (**C**) peak 1, spinacetin 3-*O*-glucosyl-(1→6)-[apiosyl(1→2)]-glucoside; (**D**) peak 2, spinacetin-3-*O*-glucosyl-(1→6)-glucoside; (**E**) peak 3, spinatoside; (**F**) peak 4, jaceidin 4′-glucuronide; (**G**) peak 5, 3′,4′,5-trihydroxy-3-methoxy-6,7-methylenedioxyflavone 4′-glucuronide; (**H**) peak 6, 5,4′-dihydroxy-3,3′-dimethoxy-6,7-methylendioxyflavone-4′-glucuronide.

**Figure 2 molecules-29-02494-f002:**
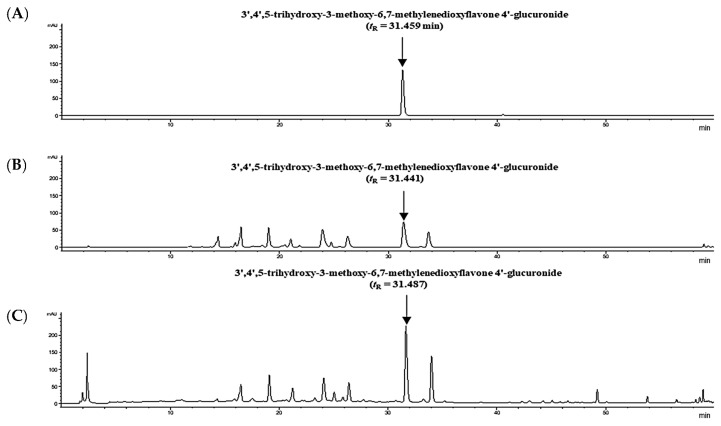
HPLC chromatogram of (**A**) the standard (TMG) solution at 62.5 μg/mL, (**B**) freeze-dried spinach (FDS), and (**C**) spinach extract concentrate (SEC) solutions at 350 nm.

**Table 1 molecules-29-02494-t001:** Characterization of flavonoid derivatives from *S. oleracea* extracts using LC-TOF/MS.

Peak	*t* _R_	Observed	Theoretical	Error	Formula	Identification
	(min)	Ion (*m*/*z*)	Ion (*m*/*z*)	(ppm)		
1	11.6	803.2242	803.2246	−0.50	C_34_H_42_O_22_	Spinacetin 3-O-glucosyl-(1→6)-[apiosyl(1→2)]-glucoside
2	12.7	671.1818	671.1823	−0.81	C_29_H_34_O_18_	Spinacetin-3-O-glucosyl-(1→6)-glucoside
3	15	523.1066	523.1087	−4.81	C_23_H_22_O_14_	Spinatoside
4	15.9	537.124	537.1244	−0.81	C_24_H_24_O_14_	Jaceidin 4′-glucuronide
5	17.7	521.0921	521.0931	−1.99	C_23_H_20_O_14_	3′,4′,5-Trihydroxy-3-methoxy-6,7-methylenedioxyflavone 4′-glucuronide
6	19.6	535.106	535.1088	−5.20	C_24_H_22_O_14_	5,4′-Dihydroxy-3,3′-dimethoxy-6,7-methylendixoyflavone-4′-glucuronide

**Table 2 molecules-29-02494-t002:** The calibration curve and responsible variables (equation, coefficient value, LOD, and LOQ) of 3′,4′,5-trihydroxy-3-methoxy-6,7-methylenedioxyflavone 4′-glucuronide (TMG) by HPLC analysis.

Repetition #	Range (μg/mL)	Regression Equation	R^2^
1	15.625–500	y = 34.065x − 33.662	0.999
2	15.625–500	y = 33.582x − 15.960	0.999
3	15.625–500	y = 33.878x − 71.495	0.999
Mean of slopes (S)	33.842	Standard deviation of intercept (σ)	23.163
LOD (3.3 × σ/S)	2.259 μg/mL		
LOQ (10 × σ/S)	6.845 μg/mL		
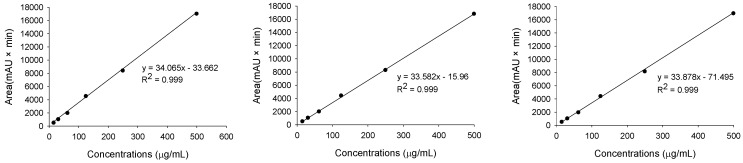			

**Table 3 molecules-29-02494-t003:** Accuracy (recovery rate) results of 3′,4′,5-trihydroxy-3-methoxy-6,7-methylene-dioxyflavone 4′-glucuronide (TMG) contents in freeze-dried spinach (FDS) solutions.

Repetition #	Spiked Amounts of TMG		
	8.98 μg/mL	17.97 μg/mL	26.95 μg/mL
1	93	101	96
2	95	101	95
3	93	101	96
Mean recovery rate (%)	94	101	96
Average recovery (%)	97		
Range of recovery rate (%)	93.3–101.0		

**Table 4 molecules-29-02494-t004:** Accuracy (recovery rate) results of 3′,4′,5-trihydroxy-3-methoxy-6,7-methylene-dioxyflavone 4′-glucuronide (TMG) contents in spinach extract concentrate (SEC) solutions.

Repetition #	Spiked Amounts of TMG		
	20.88 μg/mL	41.75 μg/mL	62.63 μg/mL
1	92	92	95
2	92	90	95
3	92	90	95
Mean recovery rate (%)	92	91	95
Average recovery (%)	93		
Range of recovery rate (%)	90–95		

**Table 5 molecules-29-02494-t005:** The intraday precision (repeatability) results of 3′,4′,5-trihydroxy-3-methoxy-6,7-methylene-dioxyflavone 4′-glucuronide (TMG) contents in freeze-dried spinach (FDS) solution.

	Amount of FDS (*n* = 5)		
	0.5 g	1.0 g	1.5 g
Repetition #	TMG Contents (μg/g)	TMG Contents (μg/g)	TMG Contents (μg/g)
1	1025	1019	1090
2	1030	994	1096
3	1028	1008	1089
4	1054	1001	1091
5	1076	956	1085
Mean	1043	996	1090
SD	19.5	21.4	3.4
RSD (%)	1.9	2.2	0.3

**Table 6 molecules-29-02494-t006:** The intraday precision (repeatability) results of 3′,4′,5-trihydroxy-3-methoxy-6,7-methylene-dioxyflavone 4′-glucuronide (TMG) contents in spinach extract concentrate (SEC) solution.

	Amount of SEC (*n* = 5)		
	0.5 g	1.0 g	1.5 g
Repetition #	TMG Contents (μg/g)	TMG Contents (μg/g)	TMG Contents (μg/g)
1	2599	2505	2455
2	2587	2464	2549
3	2531	2459	2419
4	2545	2465	2485
5	2617	2435	2466
Mean	2576	2466	2475
SD	32.5	22.5	43
RSD (%)	1.3	0.9	1.7

**Table 7 molecules-29-02494-t007:** The intraday precision (intermediate precision) results of 3′,4′,5-trihydroxy-3-methoxy-6,7-methylene-dioxyflavone 4′-glucuronide (TMG) contents in freeze-dried spinach (FDS) solution.

	Amount of FDS (*n* = 50)			
	0.5 g		1.5 g	
Day	Repetition #	TMG Contents (μg/g)	Repetition #	TMG Contents (μg/g)
1	1	1092	1	1087
	2	1104	2	1086
	3	1108	3	1089
	4	1109	4	1091
	5	1111	5	1093
2	1	1124	1	1065
	2	1109	2	1102
	3	1106	3	1090
	4	1117	4	1093
	5	1106	5	1094
3	1	1112	1	1093
	2	1111	2	1093
	3	1106	3	1096
	4	1106	4	1093
	5	1108	5	1095
4	1	1121	1	1096
	2	1121	2	1097
	3	1120	3	1096
	4	1117	4	1097
	5	1118	5	1097
5	1	1109	1	1090
	2	1103	2	1088
	3	1103	3	1089
	4	1109	4	1091
	5	1104	5	1085
	Mean	1111	Mean	1091
	SD	8.0	SD	6.7
	RSD (%)	0.7	RSD (%)	0.6

**Table 8 molecules-29-02494-t008:** The intraday precision (intermediate precision) results of 3′,4′,5-trihydroxy-3-methoxy-6,7-methylene-dioxyflavone 4′-glucuronide (TMG) contents in spinach extract concentrate (SEC) solution (*n* = 50).

	Amount of SEC			
	0.5 g		1.5 g	
Day	Repetition #	TMG Contents (μg/g)	Repetition #	TMG Contents (μg/g)
1	1	2538	1	2514
	2	2532	2	2517
	3	2532	3	2522
	4	2539	4	2571
	5	2544	5	2536
2	1	2588	1	2506
	2	2603	2	2550
	3	2588	3	2554
	4	2574	4	2556
	5	2602	5	2559
3	1	2618	1	2556
	2	2623	2	2579
	3	2638	3	2555
	4	2631	4	2623
	5	2625	5	2585
4	1	2635	1	2631
	2	2618	2	2632
	3	2613	3	2633
	4	2629	4	2636
	5	2622	5	2636
5	1	2648	1	2641
	2	2646	2	2633
	3	2628	3	2649
	4	2637	4	2655
	5	2631	5	2656
	Mean	2603	Mean	2587
	SD	37	SD	49
	RSD (%)	1.4	RSD (%)	1.9

## Data Availability

The data presented in this study are available in article and Appendix A.

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
