# Peer review of "Validation of an Analytical Method of 3′,4′,5-Trihydroxy-3-Methoxy-6,7-Methylenedioxyflavone 4′-Glucuronide for Standardization of Spinacia oleracea"

_molecules, 2024, doi:10.3390/molecules29112494_

Round 1

Reviewer 1 Report

Comments and Suggestions for Authors

Manuscript ID: molecules-2985764
Type of manuscript: Article
Title: Validation of an Analytical Methods of 3',4',5-Trihydroxy-3-Methoxy-6,7-Methylenedioxyflavone 4'-Glucuronide for Standardization of Spinacia oleracea Authors: Yun Gon Son1, Juyoung Jung2, Dong Kun Lee3, Sang Won Park4, Jeong Yoon Kim1,*, Hyun Joon Kim5,*

Special Issue Analytical Chemistry

In this article, it proposes a validation of analytical method  for standardization of 3',4',5-trihydroxy-3-methoxy-6,7-methylenedioxyflavone 4'-glucuronide (TMG) as a marker compound. For that, the specificity, accuracy, linearity, precision, repeatability, limit of detection (LOD), and limit of quantification (LOQ) have been proved according to validation guideline. The authors indicate that the specificity (for example) was confirmed from the BPI and UV chromatograms of the standard compound (TMG) in the different spinach sample. However, little further information in the introduction on the type of determination, is given. Nor is there any reference to the type of instrumental analytical technique used for identification in the title, abstract or introduction. Just a couple of lines 72-73 are used for it. Under our point of view using UPLC-QTOF/MS for the analysis gives the work a high scientific and analytical quality. But there is not much information on the state of the art, the optimisation, if any, on the methodology used or the treatment of the sample and if it has been studied.

In the same way, it is not clear which samples are to be determined and why.

For example, it has been somewhat difficult for me to find out what the abbreviations FDS and SEC stand for, given that the meaning first appears in Figure 1 and 2. The same for the acronym BPI (Based Peaks Intensity) used for confirmation of marker as UV chromatogram.

In particular, it would be highly desirable that the authors to discuss clearly the next questions:

-   Is an established analytical method used?

-   Have any instrumental parameters been studied?

-   On the basis of which criteria has the analytical methodology used been selected?

-   Is there any other methodology described? There is no literature discussion or state of the art in this respect.

Having reviewed the entire paper, in my opinion, this work has has some aspects that are novelty more related to innovation in the field of nutraceuticals than to analytical chemistry. However, there is no emphasis on that, neither in the abstract nor in the introduction.

Therefore, it is advised that the authors review the orientation of the paper and rewrite it taking into account the most relevant aspects of the paper.

Author Response

Kristina Popadić

Assistant Editor, Molecules

Dear Kristina Popadić

Thank you so much for your kind letter dated April 24, 2024, with regard to our manuscript, together with comments from the reviewers. We appreciate the hard work of the reviewers as they fairly pointed out errors and mistakes in our manuscript. We tried to revise the manuscript as much as possible in line with suggestions made by the reviewers. Corrections were colored with red. Our detailed response of reviewer’s comments was as follow.

# Response to Reviewer 1

In this article, it proposes a validation of analytical method for standardization of 3',4',5-trihydroxy-3-methoxy-6,7-methylenedioxyflavone 4'-glucuronide (TMG) as a marker compound. For that, the specificity, accuracy, linearity, precision, repeatability, limit of detection (LOD), and limit of quantification (LOQ) have been proved according to validation guideline. The authors indicate that the specificity (for example) was confirmed from the BPI and UV chromatograms of the standard compound (TMG) in the different spinach sample. However, little further information in the introduction on the type of determination, is given. Nor is there any reference to the type of instrumental analytical technique used for identification in the title, abstract or introduction. Just a couple of lines 72-73 are used for it. Under our point of view using UPLC-QTOF/MS for the analysis gives the work a high scientific and analytical quality. But there is not much information on the state of the art, the optimisation, if any, on the methodology used or the treatment of the sample and if it has been studied.

In the same way, it is not clear which samples are to be determined and why.

For example, it has been somewhat difficult for me to find out what the abbreviations FDS and SEC stand for, given that the meaning first appears in Figure 1 and 2. The same for the acronym BPI (Based Peaks Intensity) used for confirmation of marker as UV chromatogram.

→ Yes, the abbreviations were changed to full name.

In particular, it would be highly desirable that the authors to discuss clearly the next questions:

-   Is an established analytical method used?
→ This study aimed to validate the analytical method using the selected marker compound (TMG) in spinach according to AOAC international and ICH guidelines including six regulated items (specificity, LOD, LOQ, linearity, precision, and accuracy). After establishing an analytical method in this study, TMG can be used to standardize spinach sample.

-   Have any instrumental parameters been studied?
→ Instrumental parameters (HPLC) were attached in the Supplementary materials. Please see Table S1.

-   On the basis of which criteria has the analytical methodology used been selected?
→ The developed methods in this study were followed by the Codex Alimentarius Commission (CAC), and AOAC methods, including the items (specificity, accuracy, precision, quantitation limit, linearity, and range) using characteristic marker compound.

-   Is there any other methodology described? There is no literature discussion or state of the art in this respect.

→ The reason for using the specific analysis method was more clearly explained in the introduction part. Please see Line 73~80 and Line 390~392.

Having reviewed the entire paper, in my opinion, this work has has some aspects that are novelty more related to innovation in the field of nutraceuticals than to analytical chemistry. However, there is no emphasis on that, neither in the abstract nor in the introduction.

→ This study presented a method for the standardization of spinach using TMG analyzing for the development of functional foods. We have revised the abstract and introduction to emphasize the analysis methods.

Therefore, it is advised that the authors review the orientation of the paper and rewrite it taking into account the most relevant aspects of the paper.

I hope that the improved version will be acceptable for publication Molecules.

Sincerely yours.

Prof. Jeong Yoon Kim

Department of Pharmaceutical Engineering

Gyeongsang National University

Jinju 52828, Republic of Korea

[email protected]

Reviewer 2 Report

Comments and Suggestions for Authors

The novelty of the procedure should be highlighted in the introduction and conclusions sections; Is this the first report on this topic? A more detailed discussion of the literature is advised.

Specific comments:

a) Linquistic revision is required throughout the text (there are grammatical errors even in the title (... an ....methods).

b) the significant figures should be revised eg the linear range, the recoveries max (1 or better no decimal points are needed).

c) lines 64-70: Is a single compound enough for the standardization of functional foods? Some representative examples from other functional foods should be included in this part of the introduction.

d) Section 3.1: why did the authors used a single type of spinach in their research? I am pretty sure that commercially available (fresh and frozen) spinach could also have been selected for extra information.

e) Sample preparation: did the authors develop their own procedure? If not a couple of references should be added in section 3.1.

f) Section 3.4: are there any available data on the stability of the standard solution?

g) Section 3.6: a 60-min gradient to quantify a single compound using HPLC DAD at 350 nm seems to much from my point of view; the authors should justify.

Comments on the Quality of English Language

Extensive revision is required

Author Response

Kristina Popadić

Assistant Editor, Molecules

Dear Kristina Popadić

Thank you so much for your kind letter dated April 24, 2024, with regard to our manuscript, together with comments from the reviewers. We appreciate the hard work of the reviewers as they fairly pointed out errors and mistakes in our manuscript. We tried to revise the manuscript as much as possible in line with suggestions made by the reviewers. Corrections were colored with red. Our detailed response of reviewer’s comments was as follow.

# Response to Reviewer 2

The novelty of the procedure should be highlighted in the introduction and conclusions sections; Is this the first report on this topic? A more detailed discussion of the literature is advised.

Specific comments:

  1. a) Linquistic revision is required throughout the text (there are grammatical errors even in the title (... an ....methods).

→ As the reviewer’s suggestion, we will send the manuscript for English proofreading to correct grammar, if it is accepted.

  1. b) the significant figures should be revised eg the linear range, the recoveries max (1 or better no decimal points are needed).

→ As the reviewer’s suggestion, decimal points have been corrected to account for significant figures.

  1. c) lines 64-70: Is a single compound enough for the standardization of functional foods? Some representative examples from other functional foods should be included in this part of the introduction.

→ We have added the necessary marker compound establishment and test method validation according to the Korean Ministry of Food and Drug Safety guidelines to the introduction. Please see Line 73~80.

  1. d) Section 3.1: why did the authors used a single type of spinach in their research? I am pretty sure that commercially available (fresh and frozen) spinach could also have been selected for extra information.

→ In this study, we prepared two types of spinach: freeze-dried spinach using frozen spinach (FDS) and spinach extract concentrate using dried fresh spinach (SEC).

  1. e) Sample preparation: did the authors develop their own procedure? If not a couple of references should be added in section 3.1.

→ Our own procedures were used to develop spinach cultivation and preprocessing methods for nutraceutical applications.

  1. f) Section 3.4: are there any available data on the stability of the standard solution?

→ The standard solution was purchased from Sigma Aldrich and used immediately. Stability can be confirmed through the concentration-dependent peak area in the number of repetitions of sample solution and calibration.

  1. g) Section 3.6: a 60-min gradient to quantify a single compound using HPLC DAD at 350 nm seems to much from my point of view; the authors should justify. Please refer to the supplementary material for HPLC chromatograms taken under different conditions and wavelengths for comparison.

→ After analyzing spinach extract under various conditions, the condition that provided the best separation between metabolites was identified.

I hope that the improved version will be acceptable for publication Molecules.

Sincerely yours.

Prof. Jeong Yoon Kim

Department of Pharmaceutical Engineering

Gyeongsang National University

Jinju 52828, Republic of Korea

[email protected]

Reviewer 3 Report

Comments and Suggestions for Authors

The aim of this paper is to validate the quantification of a flavonoid (TMG) in Spinacia by HPLC-DAD. In the first part of the paper, six flavonoids are identified by LC and high resolution mass spectrometry. The second part of the paper validates the LC-DAD method developed for only one of the six flavonoids: trihydroxy-3-methoxy-6,7-methylendioxyflavone-4'-glucuronide (TMG).

The paper is written quite clearly and is easy to understand. However, I think the title should be clearer about the objective, which is the validation of an HPLC-DAD method. The role of LC-HRMS and LC-DAD at the beginning of the manuscript is a bit confusing.
Why only one flavonoid? Quantification of all flavonoid compounds is usually preferred in order to have a complete profile of the samples. Even if the molecules are similar, one could assume that the molar extinction coefficient is similar for all compounds and quantify the six compounds based on this hypothesis.

Why is quantification not performed by LC-MS/MS or LC-HRMS? This could be made clearer in the manuscript.

Specific comments

Abbreviations in the abstract should be avoided (FDS, SEC, BPI). No meaning is given and the abstract is supposed to be independent from the rest of the manuscript.

The sentences in 34-37 may need to be rewritten and placed in line 27, before explaining how the validation is performed.

The UV spectra obtained with DAD for the six analytes should be shown (in the paper or as supporting information). This could help to justify the 350nm wavelength used (line 82)
Why are the MS spectra acquired in positive ESI mode? Normally flavonoids are quantified in negative ESI mode.

The isotopic models could be included in supporting information to justify the identification of flavonoids (in addition to the exact mass and the mass fragments)

Line 144. Please consider rewording the sentence “but also the most sensitivity in UV…”

Section 2.3.1. Specificity should also be justified by the DAD spectra of the chromatographic peaks obtained for the Spinacia samples and the DAD spectra of the chromatographic peak of the standard. In my opinion, DAD is not very selective. So I think that it is quite likely that there are some compounds that absorb in UV and can coelute with the flavonoids. Measuring the UV spectra at different retention times of the chromatographic peak might give an indication. Specificity is also not demonstrated by the recovery studies. In this case, the flavonoid is spiked to samples that already contain the analyte. So if there is an interferent, the constant bias due to the interferent will be compensated (since the concentration found is calculated as the difference between the concentration found after and before spiking with the analyte). Specifity is easier to justify in LC-MS/MS.

Line 182. Linearity should not only be checked with the coefficient of determination. If a full validation is carried out (which is the aim of the paper) an ANOVA to verify the lack of fit should also be done. A residuals plot should also be done.
Line 190. Please consider deleting “total measurement uncertainty”. I do not see why this is written in the middle of the sentence. σ should not be used because the data provide only an estimation of the standard deviation. Strictly, s should be used in the expression.
Line 193. Please replace “trend line” with “regression line”
I think that the regression model should be calculated and plotted with the three replicates for each concentration level. The values of this overall calibration model could then be used to calculate the LOD and LOQ.
Strictly speaking, the factor 3.3 is quite approximate. If the AOAC guidelines are to be followed, the critical limit and the detection limit should be calculated. Alpha and beta probabilities should be considered.

Table 3. “Net recovery rate” should be replaced by “average recovery”. I think a t-test could be calculated to verify whether recovery is significantly different from 100%.
Line 234. Please consider rewriting the sentence with “was deviated”
Line 257. “Net RSD”. Please consider using another expression. Maybe “pooled RSD”. How is this RSD calculated? Strictly, the variances are pooled. A mean value is not calculated statistically as the mean of standard deviations or RSDs. It is the variances or RSD^2 that are pooled to give an overall value for the precision of the method.

Table 7. I think the statistics should be improved to give an estimation of the intermediate precision. In this case, ISO 5725 should be applied. From the ANOVA, the variances due to repeatibility, between-day variability and intermediate precision. It is not correct to simply calculate the standard deviation of the 25 measurements as a whole. (Same comment for Table 8).

The uncertainty of the quantification method could be calculated from the recovery and precision studies to complete the validation of the LC method.

Comments on the Quality of English Language

Minor editing is required. (see some of my previous comments)

Author Response

Kristina Popadić

Assistant Editor, Molecules

Dear Kristina Popadić

Thank you so much for your kind letter dated April 24, 2024, with regard to our manuscript, together with comments from the reviewers. We appreciate the hard work of the reviewers as they fairly pointed out errors and mistakes in our manuscript. We tried to revise the manuscript as much as possible in line with suggestions made by the reviewers. Corrections were colored with red. Our detailed response of reviewer’s comments was as follow.

# Response to Reviewer 3

The aim of this paper is to validate the quantification of a flavonoid (TMG) in Spinacia by HPLC-DAD. In the first part of the paper, six flavonoids are identified by LC and high resolution mass spectrometry. The second part of the paper validates the LC-DAD method developed for only one of the six flavonoids: trihydroxy-3-methoxy-6,7-methylendioxyflavone-4'-glucuronide (TMG).

The paper is written quite clearly and is easy to understand. However, I think the title should be clearer about the objective, which is the validation of an HPLC-DAD method. The role of LC-HRMS and LC-DAD at the beginning of the manuscript is a bit confusing.
Why only one flavonoid? Quantification of all flavonoid compounds is usually preferred in order to have a complete profile of the samples. Even if the molecules are similar, one could assume that the molar extinction coefficient is similar for all compounds and quantify the six compounds based on this hypothesis.

→ The reason for using the specific analysis method was more clearly explained in the introduction part. Please see Line 73~80 and Line 390~392. But we also presented the quantification of all metabolites in Supplementary materials. Please see Figure S7, S8, Table S2, and S3.

Why is quantification not performed by LC-MS/MS or LC-HRMS? This could be made clearer in the manuscript.

Specific comments

1. Abbreviations in the abstract should be avoided (FDS, SEC, BPI). No meaning is given and the abstract is supposed to be independent from the rest of the manuscript.

→ The abbreviations in abstract parts were revised to full name. Please see Abstract.

  1. The sentences in 34-37 may need to be rewritten and placed in line 27, before explaining how the validation is performed.

→ Abstract was resharphened more clearly. Please see Abstract.

  1. The UV spectra obtained with DAD for the six analytes should be shown (in the paper or as supporting information). This could help to justify the 350nm wavelength used (line 82)

→ As the reviewer’s suggestion, the DAD chromatogram were included in supplementary materials. Please see Figure S1 and S2.

  1. Why are the MS spectra acquired in positive ESI mode? Normally flavonoids are quantified in negative ESI mode.

→ In positive mode, the mass values of the molecular ions were well detected. It is true that flavonoid analysis in negative ion mode is common, but there are also many reports on substance analysis in positive mode. Please refer to the previous literature below.

[References]

  • Phytochemical Profile and In Vitro Bioactivities of Wild Asparagus stipularis, 2024, Molecules, https://doi.org/10.3390/molecules29040817
  • Changes in secondary metabolites in soybean (Glycine max L.) roots by salicylic acid treatment and their anti-LDL oxidation effects, 2022, Frontiers in Plant Sciences, https://doi.org/10.3389/fpls.2022.1000705
  • Antioxidant Activity, Total Phenolic and Flavonoid Content and LC–MS Profiling of Leaves Extracts of Alstonia angustiloba, 2022, Separation, https://doi.org/10.3390/separations9090234

  1. The isotopic models could be included in supporting information to justify the identification of flavonoids (in addition to the exact mass and the mass fragments)

→ In Table 1, the isotopic model was reflected. The theoretical ion displayed the same as the exact mass. The fragment pattern was described in text.

  1. Line 144. Please consider rewording the sentence “but also the most sensitivity in UV…”

→ The sentence was changed. Please see Line 158~161.

  1. Section 2.3.1. Specificity should also be justified by the DAD spectra of the chromatographic peaks obtained for the Spinacia samples and the DAD spectra of the chromatographic peak of the standard. In my opinion, DAD is not very selective. So I think that it is quite likely that there are some compounds that absorb in UV and can coelute with the flavonoids. Measuring the UV spectra at different retention times of the chromatographic peak might give an indication. Specificity is also not demonstrated by the recovery studies. In this case, the flavonoid is spiked to samples that already contain the analyte. So if there is an interferent, the constant bias due to the interferent will be compensated (since the concentration found is calculated as the difference between the concentration found after and before spiking with the analyte). Specifity is easier to justify in LC-MS/MS.

→ Although specificity could not be revealed through LC-MS/MS. However, we confirmed the separation conditions of the marker compound under various elution conditions to ensure specificity. Related information is shown in Supplementary material. Please see Fig. S1~S6.

  1. Line 182. Linearity should not only be checked with the coefficient of determination. If a full validation is carried out (which is the aim of the paper) an ANOVA to verify the lack of fit should also be done. A residuals plot should also be done.

→ All experiments that involved replicates were analyzed using both ANOVA and t-tests. The related contents were included in Section 3.11. Moreover, a residuals plot was expressed in Supplementary material. Please see Table S7 and Figure S9.

  1. Line 190. Please consider deleting “total measurement uncertainty”. I do not see why this is written in the middle of the sentence. σ should not be used because the data provide only an estimation of the standard deviation. Strictly, s should be used in the expression.

→ The sentence was removed. Please see Line 204. I understand your suggestion, but σ is indicated as LOD and LOQ derivation..

[Ref] ICH Guideline. Validation of Analytical procedures Q2(R2) ICH: Geneva, Swaziland, 2022

  1. Line 193. Please replace “trend line” with “regression line”

→ “trend line” in text was changed to “regression line”.

  1. I think that the regression model should be calculated and plotted with the three replicates for each concentration level. The values of this overall calibration model could then be used to calculate the LOD and LOQ.

→ The regression model was expressed in triplicates. Please see Table 2.

  1. Strictly speaking, the factor 3.3 is quite approximate. If the AOAC guidelines are to be followed, the critical limit and the detection limit should be calculated. Alpha and beta probabilities should be considered.

→ When calculating DL, 3.3 or 3 is usually used. Please see the attached reference for related content.

[References]

  • Development and Validation of Novel HPLC Methods for Quantitative Determination of Vitamin D3 in Tablet Dosage Form, 2024, Pharmaceuticals, https://doi.org/10.3390/ph17040505
  • Simultaneous determination of 17 regulated and non-regulated Fusarium mycotoxins co-occurring in foodstuffs by UPLC-MS/MS with solid-phase extraction, 2024, Food Chemistry, https://doi.org/10.1016/j.foodchem.2023.137624
  • Multivariate Analysis among Marker Compounds, Environmental Factors, and Fruit Quality of Schisandra chinensis at Different Locations in South Korea, 2023, Plants, https://doi.org/10.3390/plants12223877

  1. Table 3. “Net recovery rate” should be replaced by “average recovery”. I think a t-test could be calculated to verify whether recovery is significantly different from 100%.

→ “Net recovery rate” was changed to “average recovery”. Please see Table 3 and 4.

  1. Line 234. Please consider rewriting the sentence with “was deviated”

→ The sentence was rewrote. Please see Line xx.

  1. Line 257. “Net RSD”. Please consider using another expression. Maybe “pooled RSD”. How is this RSD calculated? Strictly, the variances are pooled. A mean value is not calculated statistically as the mean of standard deviations or RSDs. It is the variances or RSD^2 that are pooled to give an overall value for the precision of the method.

→ As the reviewer’s suggestion, the Net RSD was removed from the table and the notation in the text was corrected to pooled RSD.

  1. Table 7. I think the statistics should be improved to give an estimation of the intermediate precision. In this case, ISO 5725 should be applied. From the ANOVA, the variances due to repeatibility, between-day variability and intermediate precision. It is not correct to simply calculate the standard deviation of the 25 measurements as a whole. (Same comment for Table 8).

→ The results of ANOVA were expressed in Supplementary material. Please see Table S4~S8.

The uncertainty of the quantification method could be calculated from the recovery and precision studies to complete the validation of the LC method.

I hope that the improved version will be acceptable for publication Molecules.

Sincerely yours.

Prof. Jeong Yoon Kim

Department of Pharmaceutical Engineering

Gyeongsang National University

Jinju 52828, Republic of Korea

[email protected]

Round 2

Reviewer 2 Report

Comments and Suggestions for Authors

The authors have prepared a fair revision of the manuscript. I suggest acceptance of the revised version. 

Reviewer 3 Report

Comments and Suggestions for Authors

The manuscript was revised and improved according to my suggestions.

The following sentence should be rewritten: 

Page 12. Line 375. "All the experiments were conducted more than triplicates"